# Possibilities of Using Bluetooth Low Energy Beacon Technology to Locate Objects Internally: A Case Study

Jan Ližbetin *[ID] and Jan Pečman

Department of Transport and Logistics, Faculty of Technology, Institute of Technology and Business in České Budějovice, Okružní 517/10, 370 01 České Budějovice, Czech Republic; pecman@mail.vstecb.cz
* Correspondence: lizbetin@mail.vstecb.cz; Tel.: +420-387-842-190

**Abstract:** The developments that are occurring in relation to Industry 4.0 are making it possible to automate a huge number of production activities. Automation includes the possibility of automatically identifying individual elements of a system. One of the options for doing this involves the use of Bluetooth Low Energy technology. The system's advantages lie in its wide availability, economic simplicity, ability to design individual system elements, and overall system architecture. The system applied in the case study presented in this article consisted of beacons from Accent Systems and identification gateways based on the Raspberry Pi Zero W device. During several hours of testing, the functionality and reliability of all system components was demonstrated. The measurements showed that the system was able to determine the distance from a gate in line of sight with 94% accuracy. With regards to indirect visibility, when a metal crate was used to shield the beacon from the gateway, the system was able to determine the exact distance only 22% of the time. However, the variance between the actual and measured values was found to be small, therefore proving sufficient for most use cases. The major advantage of Bluetooth Low Energy beacons, and Bluetooth technology in general, is its massive ubiquity in the market. Since the Bluetooth module is part of every smartphone, this system can be made available to a wide range of users.

**Keywords:** Bluetooth Low Energy; beacon; indoor localization; automatic identification





## 1. Introduction

The ever-increasing demand for products, pressure to reduce costs, and need to deliver goods to customers as quickly as possible have been driving forces behind globalization. This trend, together with the development of technology, is contributing to the push to implement large-scale digitization. A new industrial revolution, often referred to as Industry 4.0, is currently ongoing [1,2]. In particular, the automotive industry has been undergoing rapid changes in recent years. In view of the current labor shortage, it is necessary to automate as many processes as possible, including the transfer and acquisition of information. Smart devices that can communicate with their surroundings with the help of the ubiquitous Internet enable us to continuously collect large amounts of data, the analysis and evaluation of which supports the streamlining of existing processes and the further development of society [3].

An important factor in material flow management is accurate knowledge about the movement of passive elements. For this reason, passive elements must be easily identifiable at specified points in the logistics chain, whether they are products, parts moving independently, goods in consumer packaging, or basic and derived handling and transport units. The label carrier used for identification purposes can be for a raw material or a semi-finished or finished product. If the carrier is not integral to the passive element, it must be physically bound to it, i.e., using a package, tag, label, magnetic tape, etc. According to [4,5], automatic identification is based on the use of passive and, possibly, active elements passing through the logistics chain to transfer relevant information between links in the logistics chain.

This article deals with the design and testing of a system capable of monitoring the current location of objects or people in buildings using Bluetooth Low Energy (BLE) beacon technology [6]. Such a system can be applied in many areas, for example, for monitoring current stock or the movement of handling equipment, for determining the current position of a product during production, or as a support system during the evacuation of people from buildings.

The system consisted of a gateway that captures transmitted information, and tags/beacons that transmit the information. The proposed system was tested under working conditions in an industrial setting in České Budějovice, Czechia. Testing took place according to three different scenarios, with twenty measurements taken during each.

## 2. Literature Review

Technology based on Bluetooth Low Energy is widespread. Its huge advantages for the indoor localization of objects or people are the reason this field is attracting attention in terms of research and testing, as evidenced by the number of authors currently working on the issue of localization using BLE technology and the relatively large volume of scientific studies that have already been published. The publications are devoted in part to the technology itself and the technical solutions and communication protocols that are used to transmit information, and in part to the practical use of the technology and various system tests.

The technology and methods of measuring received signals are dealt with, for example, in [7,8]. The authors state that BLE technology can be applied broadly. They propose various Bluetooth positioning methods based on received signal strength indication (RSSI), solve the problem of the indirect visibility of beacons and gateways, etc. The experiments presented in [7] showed that when using the proposed positioning method, the probability of localization accuracy up to 1.5 m was approximately 80%. The experimental results of the study in [8] showed that the maximum error during continuous positioning did not exceed 3 m.

In [9], the use of BLE technology in combination with WiFi in the interior spaces of buildings was studied. By combining these two technologies, an increase in localization accuracy was achieved.

Other improvement technologies for localization in indoor spaces using BLE technology are reported, for example, in [10–12].

Another interesting use of BLE technology is to locate moving people in objects or in business premises [13,14]. The studies do not only deal with the localization of people in buildings, but also with the use of modern smartphones, which, in cooperation with BLE technology, can provide people moving in specific places within buildings with information in real-time (e.g., navigation information, warning information, advertising information, etc.) [15,16].

The study thematically closest to the one presented in this article is [17]. The article deals with the use of BLE technology for the internal localization of various objects in an industrial setting. The authors state that in terms of the measured intensity of the RSSI signal, the results of the localization were very imprecise and susceptible to radio propagation phenomena. They concluded that high localization accuracy is only possible if the gateways are densely distributed and the propagation conditions are stable. Unfortunately, these requirements do not apply in industrial systems, where signal propagation is complex, the number of gateways receiving signals from individual beacons is limited, and propagation conditions change due to the dynamic environment. This implies that localization methods in industrial environments must take into account the complexity of the required system (e.g., maximize the number of gateways). It is obvious that the costs of acquiring and operating such a system would be higher, and this eliminates the fundamental advantages of BLE technology, which are its simplicity and low cost. In the paper, the authors present a set of localization algorithms that require relatively limited infrastructure, are not complex, and can provide valuable location information at relatively low cost. The proposed algorithms were verified under real working conditions. The localization technology was integrated

with an already existing system and was proven to work reliably despite system-level limitations and changing propagation conditions (interference, signal attenuation, and unavailability of signal measurements). The proposed approaches enabled localization in an area of 1600 m$^2$ using only 10 gateways, with an average positioning error under 8 m. The algorithms were based on signal strength measurements, which means they could also be applied to other radio technologies.

A similar study, which deals with the application of BLE technology in the construction industry, is presented in [18]. The authors developed a tracking system based on BLE technology and implemented according to the trilateration method. The aim was to obtain real-time information about the position of resources and their trajectories of movement on construction sites. The prototype was tested and implemented on a real construction site in China. The results showed that the implemented tracking system could be used on construction sites, depending on the level of accuracy required on the project.

From an analysis of available works published in prestigious scientific journals and the proceedings of scientific conferences, as indexed in the Web of Science and Scopus databases, it follows that the issue of the use of BLE technology is highly topical. In particular, the technology can be used to locate people or objects in indoor spaces. Using BLE technology, smartphones can be used to transmit various types of information, provide an overview of the movement of people around buildings, help people to navigate, or manage the evacuation of people from buildings. An equally important use of BLE technology in the industrial sector is the localization of objects, products, or property. In this context, the article adds another application in which the system delivers concrete results based on practical measurements in an industrial setting. The authors point out a possible gap in the field of research, namely, the possibility of using BLE technology in a so-called "low-cost" mode. The authors point to the possibility of implementing the technology using commonly available and low-cost components, which, after proper configuration or programming, can provide users with sufficiently reliable information. Rather than bringing a new scientific perspective on the issue, the authors point to alternative possibilities of implementation and use.

## 3. Materials and Methods

There are currently three suitable technologies on the market for the automatic identification and indoor localization of objects. These are radio frequency identification (RFID), Bluetooth Low Energy (BLE) and ultra-wideband (UWB) [19,20].

The basic principles underpinning automatic identification and object localization are the same for all three technologies, and are primarily based on two components [21]: a tag/beacon that is physically placed on the object to be tracked, and a reader to identify the tag/beacon and send data about the tag/beacon to a server, where further processing and location determination take place [22]. The tags/beacons can also be equipped with sensors for collecting telemetric data, such as temperature, humidity, pressure and vibration [23].

Despite these similarities, the terminology used to describe the aforementioned technologies differs slightly, and each handles identification and positioning in its own way.

Due to the low cost of passive tags/beacons, RFID technology is suitable for tracking large numbers of objects. The ideal application is in distribution centers or warehouses where the exact location of objects is not a requirement, and information only about the object's presence in and/or removal from the system is sufficient [24].

Due to its high accuracy and low response, UWB technology is suitable for operations where it is necessary to monitor the movement of objects or people in real time. Typical applications could be the monitoring of the movement of handling equipment or production processes [25].

BLE technology is suitable for tracking the movement of objects where high position accuracy is not of the utmost importance. Another advantage is the ability to locate beacons using mobile devices equipped with a Bluetooth module. This makes it is possible to

implement, for example, a mobile application that can help a user navigate to the place where a tracked object is located [26].

## 3.1. System Characteristics

Bluetooth is a short-range wireless technology that was developed as an alternative to a wired connection between electronic devices. Today, Bluetooth is found in practically every electronic device, including mobile phones, tablets, computers, wearable electronics such as smart watches, headphones, and sensors that monitor health functions. Bluetooth is undoubtedly among the most widespread technologies on the planet. In 2019, according to the organization Bluetooth SIG, more than 4.2 billion manufactured electronic devices featured this technology, and this is predicted to reach 6.2 billion devices by 2024 [27,28].

Part of the proposed system for the indoor localization of objects was a gateway for recognizing beacons, which requires programming accordingly. The first question that needed to be answered was which microcontroller or microcomputer to choose for this purpose. The requirements were simple: the device had to have a Wi-Fi module for communication with the application programming interface (API) [29], and a Bluetooth module that enables scanning. From the conducted research, it emerged that two devices met these requirements: the ESP 32 and the Raspberry Pi Zero W. Both devices can be purchased in the price range of $10 to $15 each [30]. The Raspberry Pi Zero W also requires a memory card on which the operating system is stored and to which the scripts are subsequently uploaded. A more detailed description of both devices, including the programming thereof, is given in [31]. After programming both devices, the Raspberry Pi Zero W device was selected for testing due to its memory capacity.

When choosing tags/beacons, products from Estimote, Gimbal, Kon-takt.io, Quppa, Accent Systems, and others were considered. The ideal beacon had to meet the following criteria:

- Two-year battery life;
- Compact dimensions;
- Range of at least 30 m;
- iBeacon and Eddystone beacon protocol support;
- Low cost.

Although the products of several companies met the defined criteria, the portfolio of beacons offered by Accent Systems seemed ideal for the proposed system. The decisive factor was not the price, even though it was more favorable than the competition, but the accompanying technical documentation, which included measured range values at different transmission powers. Since the beacons were to be subjected to their own measurements followed by a comparison with the values from the manufacturer, we had to choose a manufacturer that provided this data [32,33].

The models of the beacons offered covered all use cases that a customer implementing such technology could use. A comparison of the models according to the identified criteria is presented in Table 1.

**Table 1.** Accent Systems beacon parameters.

|  | **iBKS PLUS** | **iBKS 105** | **iBKS CARD** | **iBKS USB** |
|---|---|---|---|---|
| Battery life at 1 s transmission interval (months) | 120 | 40 | 18 | ∞ |
| Price (USD) | 22.30 | 12.30 | 16.20 | 13.60 |
| Range (m) | 100 | 50 | 100 | 100 |
| Dimensions (mm) | 24 × 84 × 84 | 11.3 × Ø52.6 | 54 × 85 × 4.3 | 5.1 × 18.9 × 38 |

Source: https://accent-systems.com/buy-beacons/, accessed on 12 June 2021.

Based on the calculation according to the scoring method, described in detail in [31], the iBKS 105 model was selected. The iBKS 105 beacon (see Figure 1) was not only the best

fit, except in the weaker range (50 m) (although this is sufficient for indoor use), but also could be retrofitted with an accelerometer.

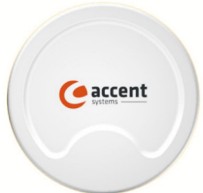

**Figure 1.** iBKS 105 beacon (Source: https://accent-systems.com/product/ibks-105/, (accessed on 12 June 2021).

### 3.2. System Architecture

The system architecture for the indoor localization of objects, as shown in Figure 2, consists of several components that communicate with one another. These components are the beacons, which move around in various ways, and the gateways, which detect the beacons and send the collected data for validation and storage in a database.

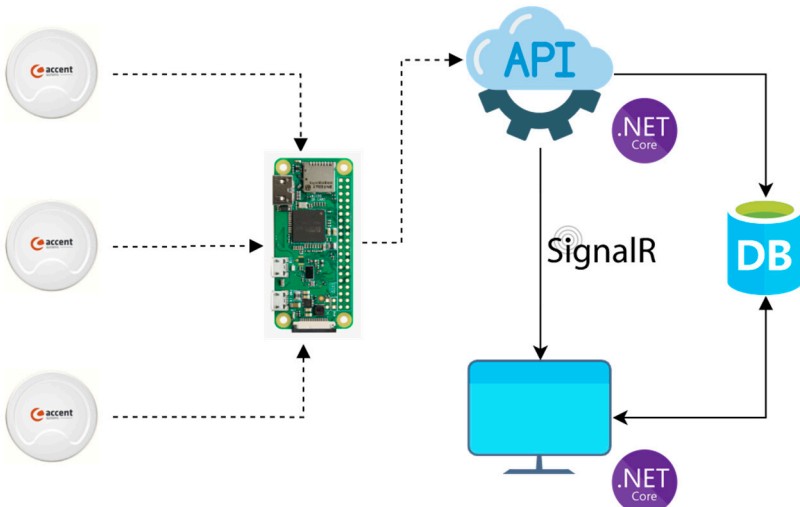

**Figure 2.** System architecture.

A web client enables users to monitor the location of the beacons as well as the flow of requests [34,35]. The basic assumptions are that the beacons are placed within range of the gateway, and the data needed to visualize the current location of the beacons is loaded directly from the database by the web client [36,37].

The transmission power of the iBKS 105 beacon was set to −4 dBm and the time period during which the beacon sends a broadcast packet was set to 1000 ms. By setting the transmission power to −4 dBm, which had no impact on the beacon's use for measurement purposes, the battery life was extended by ten months compared to if a setting of +4 dBm, and by three months compared to a setting of 0 dBm.

The design of the database (see Figure 3) model was approached such that the resulting software could be used by multiple companies, i.e., so that each company could manage and display their own beacons. Each company stored in the database can have multiple buildings around the world. For each building, the GPS coordinates of longitude and latitude are stored in the table. A building can also have multiple floors, which are stored in the Floor table, the most interesting column of which is the Map column, which stores an image with the layout of the entire floor. Rooms, or sections, are stored in the Rooms table, which also stores its internal location for each record. With the help of this information, it is possible to visualize which rooms or sections are available on the given map. The indoor position is stored in the IndoorCoordinate table, which contains columns to store

four coordinates. The last location-related part is the Gateway table, which stores data on the available scanning gateways, with each tied to a certain room, and with the coordinates of each gateway being part of the table. There can also be several gateways in one room or section, which can then be visualized on the map according to the available coordinates.

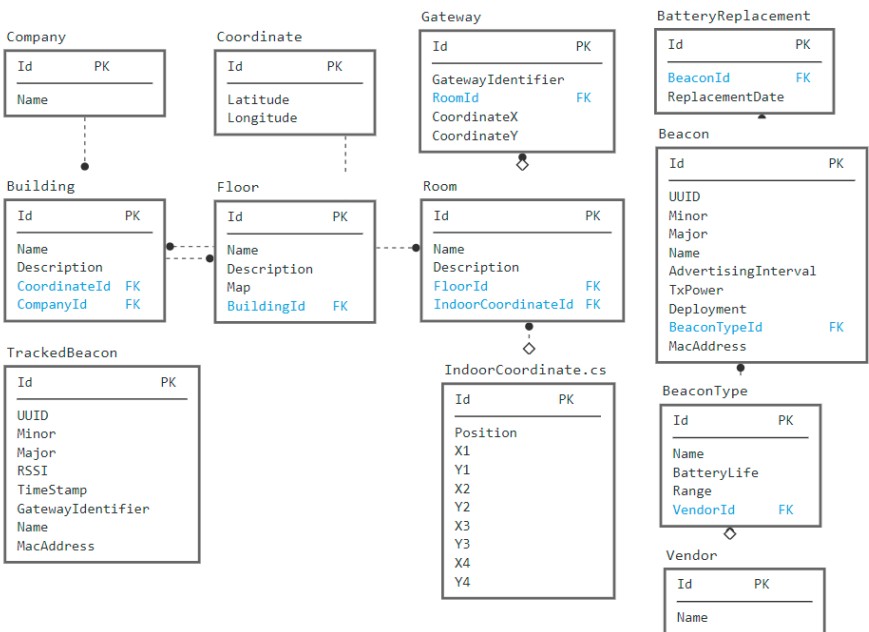

**Figure 3.** Database model.

The second logical unit relates to the management of the beacons themselves. The beacons are stored in the Beacon table, which mainly contains information about their configuration. The model of beacon is stored in the BeaconType table, which includes data on the manufacturer, range declared by the manufacturer and battery life. The BatteryReplacement table is closely related to battery life, and contains data on when and in which beacon the battery was replaced. In combination with the data on when the beacon was first installed in the Deployment column of the Beacon table and the BatteryLife data from the BeaconType table, it is then possible to notify the user that the time to replace the battery is approaching [38,39].

The last logical unit relates to the recording of beacons detected by Bluetooth gateways in the TrackedBeacon table. Each entry in this table contains a beacon identifier, a Bluetooth gateway identifier, and an RSSI value to evaluate the section or room in which the beacon is located [40].

Another essential component of the system architecture is a web client, which enables monitoring of the current number of beacons based on the selected floor map. The web client also enables monitoring of the flow of requests from Bluetooth gateways to the API.

The tracking module allows the user to select a company from the database. This subsequently populates the building drop-down list, from which one specific building can be selected. After clicking on the desired building, the floor drop-down list appears. By clicking on the required floor and then the "Load map" button, a floor map based on the specified filter is presented. By clicking on the "Load beacons" button, the Bluetooth gateways located on the selected floor are marked on the map. The locations of the gateways are marked with a circle based on the coordinates in the database. If there is no beacon within range of the gateway, the circle is red; if there is at least one beacon nearby, the color of the circle is orange; and if there are two or more beacons nearby, the circle is green. After clicking on a specific circle, a label pops up with the name of the selected gateway and the number of beacons in its vicinity. In addition to marking the gateways on the map, a table

containing a list of the individual beacons located in the given section is displayed on the right side of the screen.

The scanning module enables live monitoring of the flow of requests from Bluetooth gateways to the API by establishing a connection to the API. If the connection is successfully established, the data directed to the API is constantly updated in a table, without the need to refresh the browser window. This module is suitable for verifying the functionality of Bluetooth gateways and generally monitoring the data the gateways send to the API.

## 4. Results

The system was tested under working conditions in an industrial setting in České Budějovice, Czechia. The hall in which the testing took place is shown schematically in Figure 4.

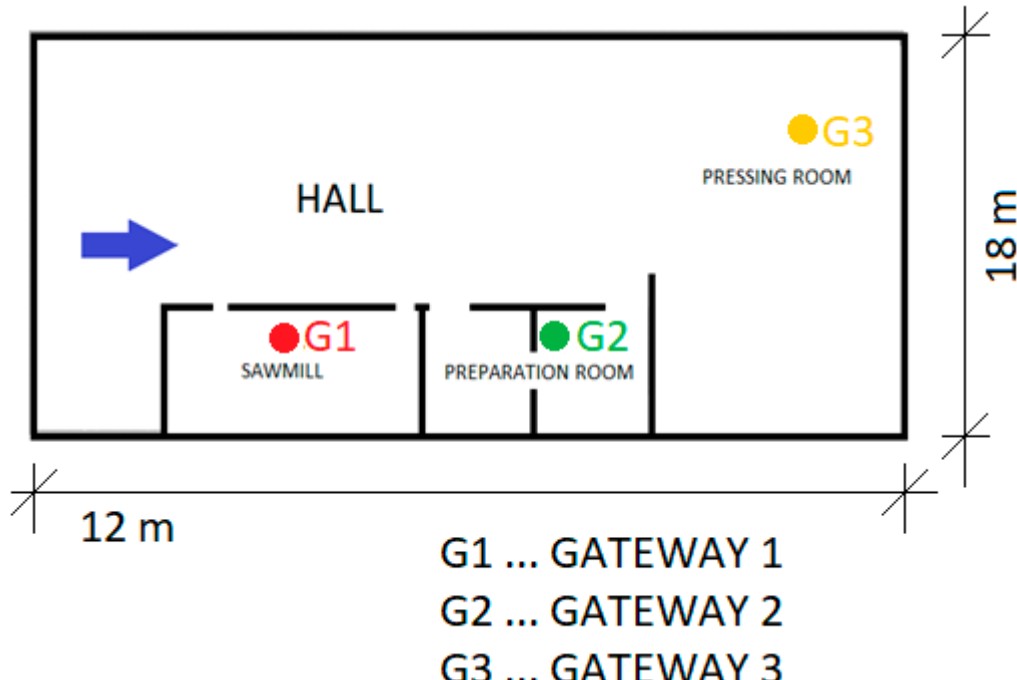

**Figure 4.** Hall layout.

Before actual testing was begun, it was necessary to provide power to all the gateways, connect to the Internet, and verify the functionality of the connection with the web API. Power was temporarily provided to the gateways using power banks. The second web application module, which displays the current traffic of all data directed to the web API, was ideal for functionality verification. This module made functionality verification simple and fast.

The hall was divided into three sections in which gateways were placed for detecting the presence of beacons. The first gateway was located in the "Pressing Room", the second in the "Preparation Room" (see Figure 5), and the last in the "Sawmill". The gates were placed as close to the middle of the sections as possible.

The testing consisted of attaching a beacon to the side of a metal crate (see Figure 6) placed on a cart. Since there were only three beacons available, only three crates were prepared in this way. The carts were subsequently moved between the mentioned sections to check whether the system responded to changes in position. The distance between the gateway and the crate was always 3 m. A variant without shielding, where there was no obstacle between the crate and the gateway, was tested, as was a variant with shielding, for which the crate was turned 180 degrees, thereby providing shielding itself.

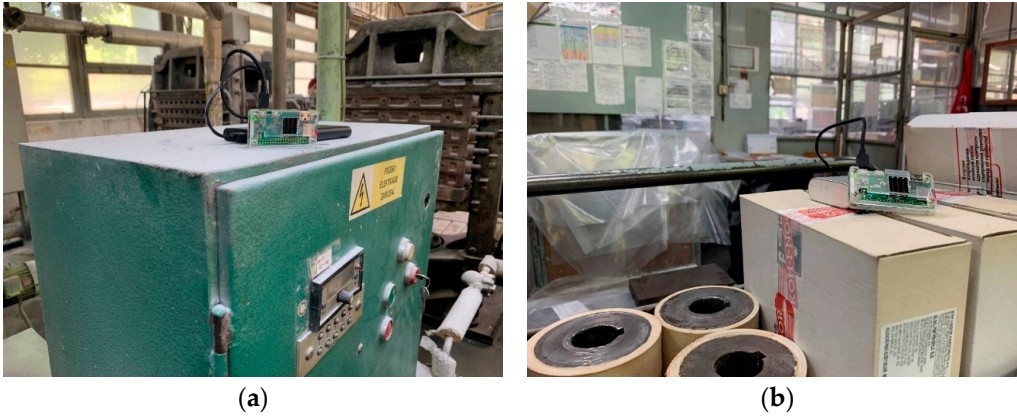

| (**a**) | (**b**) |

**Figure 5.** Placement of gateways in: (**a**) Pressing Room; (**b**) Preparation Room.

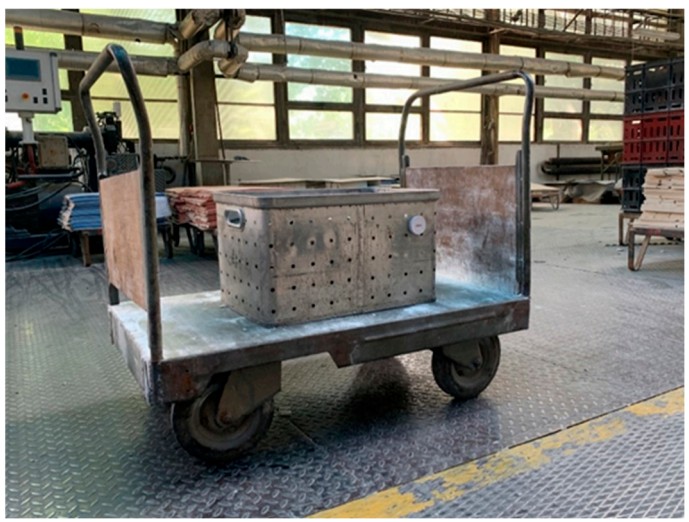

**Figure 6.** Crate equipped with iBKS 105 beacon.

The following scenarios were established to check how successful the individual gateways were in detecting the beacons:

1. Placement of one crate in each section in which a gateway was placed.
2. Placement of two crates in the Preparation Room and one in the Press Room.
3. Placement of two crates in the Preparation Room and one in the Sawmill.

Regarding the "success rate," the measurement was considered successful if the object was detected at a distance of at least 5 m, or if the difference between the actual distance and the detected distance did not exceed 5 m.

In all, 20 measurements were taken for each scenario and gateway, the results of which are presented in Tables 2–5.

**Table 2.** Scenario 1: Success rate of determining distance from gateways.

| | Scenario 1 | | | | | |
|---|---|---|---|---|---|---|
| **Distance** | **3 m** | | **5 m** | | **10 m** | |
| **Location** | **Without Shielding** | **With Shielding** | **Without Shielding** | **With Shielding** | **Without Shielding** | **With Shielding** |
| Pressing Room | 95% | 25% | 5% | 60% | 0% | 15% |
| Preparation Room | 90% | 20% | 10% | 55% | 0% | 25% |
| Sawmill | 95% | 25% | 5% | 50% | 0% | 25% |

**Table 3.** Scenario 2: Success rate of determining distance from gateways.

| | Scenario 2 | | | | | |
|---|---|---|---|---|---|---|
| **Distance** | **3 m** | | **5 m** | | **10 m** | |
| **Location** | **Without Shielding** | **With Shielding** | **Without Shielding** | **With Shielding** | **Without Shielding** | **With Shielding** |
| Pressing room | 95% | 20% | 5% | 55% | 0% | 25% |
| Preparation room | 92.5% | 25% | 7.5% | 57.5% | 0% | 17.5% |

**Table 4.** Scenario 3: Success rate of determining distance from gateways.

| | Scenario 3 | | | | | |
|---|---|---|---|---|---|---|
| **Distance** | **3 m** | | **5 m** | | **10 m** | |
| **Location** | **Without Shielding** | **With Shielding** | **Without Shielding** | **With Shielding** | **Without Shielding** | **With Shielding** |
| Preparation room | 92.5% | 17.5% | 7.5% | 62.5% | 0% | 20% |
| Sawmill | 95% | 20% | 5% | 60% | 0% | 20% |

**Table 5.** Total success rate of distance determination.

| | Total Success | | | | | |
|---|---|---|---|---|---|---|
| **Distance** | **3 m** | | **5 m** | | **10 m** | |
| **Location** | **Without Shielding** | **With Shielding** | **Without Shielding** | **With Shielding** | **Without Shielding** | **With Shielding** |
| Total | 94% | 22% | 6% | 57% | 0% | 21% |

Based on the measurements, it was found that in the case of direct visibility, the system was able to determine the distance with an accuracy of 94%. In 6% of cases, the system indicated a distance of 5 m compared to the actual distance of 3 m. Furthermore, the measurements confirmed that the reduced signal strength was caused by shielding. For the variant with shielding, the measurement results were significantly worse, with the system only able to determine the correct distance in 22% of cases. In 57% of cases, the determined distance was 2 m longer than the actual distance, and in 21% of cases the distance was off by as much as 7 m.

If at a given moment the RSSI value was only slightly lower than the extreme limit of the reference value of 3 m, the system would show the second closest distance, which in this case was 5 m. The same applied for a distance of 5 m, with the system showing a distance of 10 m.

To determine the position more precisely, it would be necessary to carry out more detailed measurements, which would be graduated in one-meter increments. This measurement would be worth considering at least up to a distance of 10 m. Due to signal fluctuations caused by the environment in which the beacons were located and the already small spread of RSSI values at given distances, the accuracy of the measurements was not completely reliable. For this reason, the approximate distance can be consideredinformational.

A visualization of Scenario 2 with two crates is shown in Figure 7, which is a screenshot of the web interface of the application. The system correctly recognized the number of crates in the individual sections and marked the gateway in red where there was no crate. The gateway with two crates within range is also accurately marked green. In the tables for the individual gateways, it is possible to monitor more detailed data about the scanned beacons.

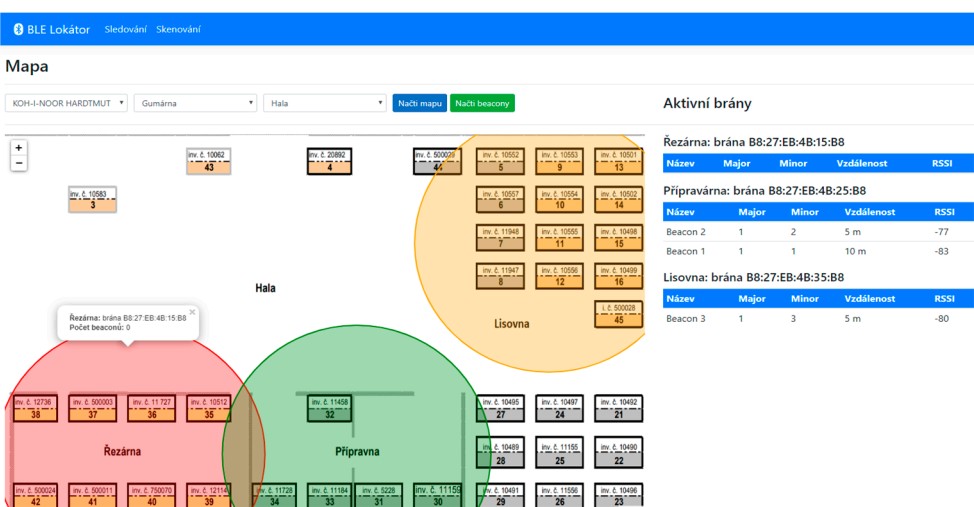

**Figure 7.** Tracking module testing (screenshot of web interface screen).

## 5. Discussion

The designed system and all its subcomponents were subjected to a load test for several hours. The gateways sent requests and collected data at regular intervals about available beacons in the vicinity of the API [41]. It then processed the requests and stored them in the database. With the support of the WebSocket protocol and the connection between the API and the web client, it was possible to monitor the flow of data from the individual gateways directly in the web application, without the need to update the browser window. In the web application, it was also possible to monitor the number of beacons in the individual sections in which the gateways were located. During testing, no problems occurred that caused the measurements to be interrupted. The functionality of all system components was demonstrated [42,43].

During the measurements, the signal strength was attenuated by shielding the beacon on the crate wall from the gateway. This had a negative effect on the indicated distance of the beacon from the gateway. However, this is a feature known to affect BLE beacons, which is why the use of this technology not considered suitable if highly accurate location determination is required. UWB technology is more suitable in such cases.

The measurement itself showed that the system was able to determine the distance from the gate with 94% accuracy in line of sight. In indirect visibility, when there was shielding by a metal crate between the beacon and the gate, the system was able to determine the exact distance only 22% of the time. However, the variance between the actual and measured value is relatively small, and is sufficient for most use cases, since the goal of these measurements was to confirm the use of technology to locate objects in the internal spaces of business that have relatively limited (small) dimensions. It is therefore not necessary to define the exact location of an object, but only the part of the hall in which the given object is located. When the signal was shielded, the object was detected at a distance of up to 5 m in 57% of measurement cases, which we considered acceptable.

Based on the testing, the system consistently demonstrated the ability to determine the section in which the given beacon was located. As there was only one gateway in each section, it was not possible to accurately determine on the map whether the crate was located north, east, south or west of the gateway. In order to determine the location of the beacon more precisely, it would be necessary to use at least three gateways for each section to serve as reference points, which would have to be located within reach of each other. Subsequently, the approximate position could be determined using the triangulation method [44]. This approach would make the system considerably more expensive [45,46], and there would also be increased infrastructure costs due to the increased number of requests directed to the API.

## 6. Conclusions

The aim of the article was to characterize and test a system capable of locating objects inside buildings using BLE beacon technology.

In the first step, the authors conducted a survey of available devices (hardware) and proposed a low-cost combination of system elements. The next step was the configuration of the system and the programming of reading gates and web applications, which are used to monitor the flow of data and thus the location of individual elements. The last step was system testing. The system was tested at a manufacturing company in Če­ské Budějovice that provided a dedicated space for testing free of charge. The conditions were therefore specifically set for the hall in which the functionality was tested and in which the system could theoretically be deployed later.

During several hours of testing, the functionality and reliability of all system components was demonstrated. The measurements showed that the system was capable of determining the distance from the gateway with 94% accuracy in line of sight. When a metal crate was used to shield the beacon from the gateway, the system was able to determine the exact distance only 22% of the time.

Based on these results, it is clear that BLE beacon technology can be used to determine the location of objects in buildings, industrial halls or warehouses. However, the accuracy of this technology is not as high in cases of indirect visibility, which can be caused by shielding between the beacon and the gateway. For this reason, it is necessary to take the indicated distance as informational. However, the variance between the actual and measured values was found to be small, therefore proving sufficient for most use cases.

On the basis of the testing and the obtained results, it is possible to define the verified advantages and disadvantages of the system. The main advantage of the tested system is the low cost; the devices are available in the range of USD 5 for a beacon and about USD 15 for a Raspberry Pi Zero W micro-computer. The disadvantage of the system is that the data obtained is only indicative and does not provide an exact location. However, this was not the aim of the research. The goal was to identify the part of the hall in which the object was located. A major advantage of BLE beacons, and Bluetooth technology in general, is its ubiquity in the market. Since the Bluetooth module is part of every smartphone, further development of the system using mobile readers will be much cheaper than for specialized RFID readers or devices with a UWB module.

In conclusion, the authors would like to outline another direction of research that could refine the location of objects. Obviously, this technology has not been verified in all possible scenarios and environments, as the signal can be jammed. However, the goal of the research was not to cover all scenarios, but to verify the functionality of the system designed and configured at low cost. The authors therefore suggest further possible development in wider and more detailed testing of the system, also taking into account other factors influencing signal reception and thus the location of objects.

**Author Contributions:** Conceptualization, J.L.; methodology, J.L.; validation, J.L. and J.P.; formal analysis, J.P.; data curation, J.L.; writing—original draft preparation, J.L.; writing—review and editing, J.P.; visualization, J.L.; supervision, J.P. All authors have read and agreed to the published version of the manuscript.

**Funding:** This research received no external funding. The company where the operation of the system was tested provided its premises free of charge.

**Institutional Review Board Statement:** Not applicable.

**Informed Consent Statement:** Not applicable.

**Data Availability Statement:** Not applicable.

**Acknowledgments:** This article was prepared with funding from the Ministry of Education, Youth and Sports of the Czech Republic, grant 05SVV2302, "Proposal of methodology in the context of investigating the influence of the height profile of roads on the reduction of emissions from road transport".

**Conflicts of Interest:** The authors declare no conflict of interest.

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
