# Peer review of "Possibilities of Using Bluetooth Low Energy Beacon Technology to Locate Objects Internally: A Case Study"

_technologies, doi:10.3390/technologies11020057_

Round 1

Reviewer 1 Report

Dear Authors,

Thank you for sending your research paper to the journal Technologies. Your paper is interesting and within the scope of the journal.

Bluetooth applications are an interesting filed, especially in closed spaces. Also, your approach to a functional low-cost solution is excellent.

I would like to propose you some improvements to the current version of your paper:

1.       Can you state at the end of the paper that your research is or is not paid for by the companies that you used in your research? This is an important aspect of validating the results.

2.       Try to not use so high amount of so-called chain citations. For example [2-4], each reference is valued at least one sentence.

3.       In the last paragraph of the Literature review can you state a clear research gap?

4.       Can you define the term “low price”?

5.       Also, maybe you can use the term “low-cost solution”?

6.       In the abstract you stated: “the variance 21 between the actual and measured values was found to be small, therefore proving sufficient for most 22 use cases”. Can you comment on this in the chapter Discussion? It will be interesting to see full statistics.

7.       The conclusion is pretty generic. Please add your methodology, a short explanation of the experiment, the advantages and disadvantages of your research, and maybe the next research steps (for example financial analysis).

Regards,

Author Response

We would like to thank the reviewer for his insight into the subject addressed.

  1. Can you state at the end of the paper that your research is or is not paid for by the companies that you used in your research? This is an important aspect of validating the results.

Information has been added. The company provided the space for testing free of charge.

  1. Try to not use so high amount of so-called chain citations. For example [2-4], each reference is valued at least one sentence.

Citations have been edited.

  1. In the last paragraph of the Literature review can you state a clear research gap?

The Literature review chapter was expanded and supplemented.

  1. Can you define the term “low price”? Also, maybe you can use the term “low-cost solution”?

We agree with the reviewer's opinion. We replaced the term "low price" with the term "low-cost solution".

  1. In the abstract you stated: “the variance between the actual and measured values was found to be small, therefore proving sufficient for most 22% use cases”. Can you comment on this in the chapter Discussion? It will be interesting to see full statistics.

We expanded and supplemented the discussion.

  1. The conclusion is pretty generic. Please add your methodology, a short explanation of the experiment, the advantages and disadvantages of your research, and maybe the next research steps (for example financial analysis).

We supplemented the conclusion with a brief methodology, a description of testing, we defined the advantages and disadvantages of the system, and we also added an opinion on further possible research, or further system testing.

Reviewer 2 Report

It is not very clear how to interpret this work. It shows that the BLE scanning technology is working. Yes it is. There can be many obstacles to the propagation of a radio signal, and all of them were not explicitly considered in the work. What is the scientific component of this work?

Author Response

We would like to thank the reviewer for his insight into the subject addressed.

It is not very clear how to interpret this work. It shows that the BLE scanning technology is working. Yes it is. There can be many obstacles to the propagation of a radio signal, and all of them were not explicitly considered in the work. What is the scientific component of this work?

We have to agree with the reviewer's opinion. We are aware that not all factors that influence the measured signal, or may cause signal interference. However, this was not even the goal of the testing. The testing took place in a company that allowed us to perform measurements in their specific conditions free of charge, and the main goal was to verify the functionality of the proposed system, which was built on a so-called "low-cost" solution. The goal was to verify the functionality of the programmed gateway, cooperation with the programmed application, which we succeeded in doing.

The article is presented as a case study and verification of the proposed system. The contribution of the article lies in the alternative design of the system solution in a low-cost mode, where individual components are commonly available in a price range of up to several tens of dollars. For clarity, we incorporated these statements into the text of the article, where the Discussion and Conclusion chapters were modified.

Reviewer 3 Report

Some parts of the text need to be improved:

1) In Fig.4, it will be better if the locations of the gateways are displayed.

2) In Fig. 4, the size of the hall may be described. 

3) Before presenting the results in Table 2,3, and 4, the definition of 'success rate' is described in detail. 

Author Response

We would like to thank the reviewer for his insight into the subject addressed.

  1. In Fig.4, it will be better if the locations of the gateways are displayed.

The image has been modified.

  1. In Fig. 4, the size of the hall may be described.

Image has been added.

  1. Before presenting the results in Table 2,3, and 4, the definition of 'success rate' is described in detail.

The "success rates" has been defined and incorporated into the text.

Round 2

Reviewer 1 Report

Dear Authors,

Thank you for the updated version of your paper.

You have answered all my raised questions.

Regards,

Reviewer 2 Report

The authors expressly indicated in the text of the work that they did not pursue scientific goals. Their purpose is to demonstrate how the system works. This has been demonstrated. If this complies with the policy of the journal, then you can publish such an article.